# DBS for Obesity

**DOI:** 10.3390/brainsci6030021

**Published:** 2016-07-18

**Authors:** Ruth Franco, Erich T. Fonoff, Pedro Alvarenga, Antonio Carlos Lopes, Euripides C. Miguel, Manoel J. Teixeira, Durval Damiani, Clement Hamani

**Affiliations:** 1Division of Pediatric Endocrinology, Children’s Hospital, University of São Paulo Medical School, São Paulo 05403-000, Brazil; franco.ruth@icloud.com (R.F.); durvald@iconet.com.br (D.D.); 2Division of Functional Neurosurgery of Institute of Psychiatry, Department of Neurology, University of São Paulo Medical School, São Paulo 01060-970, Brazil; erich.fonoff@gmail.com (E.T.F.); manoeljacobsen@gmail.com (M.J.T.); 3Department of Psychiatry, Institute of Psychiatry, University of São Paulo Medical School, São Paulo 01060-970, Brazil; pedrodealvarenga@gmail.com (P.A.); antonioclopesmd@gmail.com (A.C.L.); ecmiguel7@gmail.com (E.C.M.); 4Division of Neurosurgery, Toronto Western Hospital, University of Toronto, Toronto, ON M5T 1R8, Canada; 5Division of Neuroimaging, Centre for Addiction and Mental Health, Toronto, ON M5T 1R8, Canada

**Keywords:** deep brain stimulation, obesity, hypothalamus, nucleus accumbens

## Abstract

Obesity is a chronic, progressive and prevalent disorder. Morbid obesity, in particular, is associated with numerous comorbidities and early mortality. In patients with morbid obesity, pharmacological and behavioral approaches often have limited results. Bariatric surgery is quite effective but is associated with operative failures and a non-negligible incidence of side effects. In the last decades, deep brain stimulation (DBS) has been investigated as a neurosurgical modality to treat various neuropsychiatric disorders. In this article we review the rationale for selecting different brain targets, surgical results and future perspectives for the use of DBS in medically refractory obesity.

## 1. Introduction

Obesity is a chronic and progressive disorder with a prevalence of 600 million individuals worldwide [1]. In 2014, approximately 39% of the adult population was considered to be overweight and 13% obese [1]. Unfortunately, this number is on the rise [2]. Such an increased incidence is problematic due to the associated comorbidity and the reduced life expectancy in patients bearing the disease [3].

Morbid obesity is defined as a body mass index (mass/height^2^) >40 kg/m^2^. It affects more than eight million Americans with a prevalence of approximately 14% [4]. Morbidly obese patients not only die prematurely, but they also have a poor quality of life [5]. This is due in part to the numerous co-morbidities associated with the disease, including diabetes, cardiovascular disorders, osteoarthritis, hepatic steatosis, among others.

A major problem in patients with morbid obesity is that pharmacological and behavioral approaches often have limited results [6,7]. Surgical interventions, including bariatric procedures, are currently being used with variable outcomes and a non-negligible incidence of side effects [8]. That said, bariatric surgery is currently the most efficacious treatment for rapid weight loss in morbid obesity, with overall clinical results superior to those achieved with the best medical management [9,10]. In addition to side effects, a common problem with the bariatric surgery is the relatively high incidence of recurrence [11]. In fact, recent long-term follow-up studies have shown that up to 46% of patients may regain weight in the postoperative period [12]. One of the main factors associated with disease recurrence is compulsive eating [13,14]. It has now been suggested that patients who present binge eating disorders or loss of control eating have less weight loss and/or more weight regain after bariatric surgery [14]. This stresses the fact that obesity cannot be simply regarded as an endocrinological condition, but as a disease with a strong neuropsychiatric component.

Deep brain stimulation (DBS) involves the delivery of electrical current to the brain parenchyma. This is accomplished by implanting electrodes into specific brain targets and connecting them to a pacemaker (i.e., implantable pulse generator) [15]. The latter is programmed so that current may be delivered at different amplitudes, pulse widths and frequencies in monopolar or bipolar configurations. The electrodes most commonly used today have four different contacts, which may be activated alone or in combination. Depending on the stimulation parameters and brain target, different neural elements or circuits may be involved in a DBS response [16]. In recent years, a few reports have been published using DBS to treat obesity [17,18,19]. In this article we review the rationale for selecting different brain targets, surgical results and future perspectives of using DBS for treating medically refractory obesity.

## 2. Anatomical Targets

The pathophysiology of obesity involves not only altered patterns of eating and satiety but also reward and compulsive aspects of food intake. As such, DBS targets currently proposed to treat obesity include the hypothalamus and nucleus accumbens (NAc) [20,21].

The hypothalamus may be subdivided in various anatomical and functional subregions/nuclei. Some of the most commonly involved in mechanisms of feeding and energy balance are the arcuate nucleus (ARC), the dorsal medial nucleus, the paraventricular nucleus, the lateral hypothalamus (LH) and the ventral medial nucleus (VM). An in-depth review of the neurocircuitry of feeding and satiety, including all the cell types and peptides involved, may be found elsewhere [22,23,24]. Of particular interest are the VM and LH, as these nuclei are being considered as potential targets for DBS surgery. The VM is a relatively large nucleus with an abundance of leptin receptors [22]. Leptin and insulin provide the hypothalamus with peripheral signals of adiposity [25]. If levels of these hormones are high, the organism reduces feeding. In the hypothalamus, prominent levels of leptin and insulin receptors may also be found in ARC [22,24]. In this nucleus, these hormones modulate activity of populations of cells expressing neuropeptide Y/agouti gene-related protein and cocaine- and amphetamine-related transcript (CART)/pro-opiomelanocortin (POMC) [22,26]. In general, states of negative energy deficit (e.g., starvation) increase the activity of ARC NPY/AGRP neurons, ultimately favoring food consumption. Overall, ARC NPY/AGRP neurons project to most hypothalamic nuclei involved in feeding control [22].

The LH extends through most of the anteropoesterior axis of the hypothalamus [27]. It is composed of diffuse populations of neurons intermingled with fibers, largely from the medial forebrain bundle (MFB) [27]. Subpopulations of cells in LH express different peptides and hormones, including orexins and melanin-concentrating hormone (MCH). Both have orexigenic effects and tend to increase food consumption [22,27].

In contrast to the hypothalamus, the NAc has been suggested to play a role in rewarding aspects of food intake and compulsive feeding. Some of the evidence suggesting an involvement of the NAc in the pathophysiology of obesity includes the following: the pattern of compulsive eating shown in some forms of clinical and preclinical obesity often resembles that of drug addiction [28,29,30]. Food craving and the anticipation of food reward in preclinical models are associated with changes in D2 striatal receptors [31]. In rodents, binging on sugar and the ingestion of fat diets increase the release of dopamine in the NAc [32,33]. In addition to the above-mentioned evidence, numerous neuroimaging studies using positron emission tomography or functional magnetic resonance imaging have been conducted in obese patients at baseline and during activation tasks (for a review see [34,35]). Overall, sensory stimuli related to palatable foods seem to activate cortical regions and the reward circuitry, including the ventral striatum [34,35]. In some studies, hyper-responsiveness of reward-related regions has been suggested to forecast a poor outcome to weight-loss programs [36]. Though not a consensus, studies in obese patients treated with bariatric surgery have shown that the response of the ventral striatum to images of caloric food is less pronounced than that recorded prior to surgery [37]. Also commonly investigated with neuroimaging is the status of the dopaminergic system. Though results are not consistent across trials, a few studies have shown that food-related sensory stimuli elicit dopamine release and that obese individuals have a low D2 binding in the striatum [34,35,38,39,40].

## 3. Preclinical Studies

Based on studies in which lesions, pharmacological agents and electrical stimulation were applied to the hypothalamus, as well as on clinical cases of patients with brain tumors, the VM and LH have been suggested as being “satiety” and “feeding” hypothalamic centers, respectively [41,42,43]. Though this “dual center hypothesis” is somewhat outdated, much has been learned from experiments manipulating hypothalamic regions to investigate mechanisms and neurocircuits of food intake and satiety. Early preclinical work in which either the VM or LH were lesioned or pharmacologically inhibited has shown an increase in feeding or satiety, respectively [44,45]. In contrast to the relative uniformity of conclusions reached by the above-mentioned studies, results with the use of electrical stimulation are far more complex. As the hypothalamus is involved in numerous physiological functions, stimulation of different nuclei may influence multiple physiologic processes. In addition, current may spill over and modulate activity in adjacent nuclei and nearby structures, including the fornix and medial forebrain bundle. Stimulation delivered to the VM of rodents [46,47,48,49], dogs [50], mini-pigs [51], and nonhuman primates [52] at settings known to drive neurons and axonal projections has been shown to alter feeding behavior, the type of food ingested and/or has slowed down weight gain over time. However, these results are not consistent with a few studies in these same species showing that stimulation may not be effective [53] or even increase food consumption [54]. Though part of the discrepancies across studies may be explained by the use of different stimulation settings (e.g., 50 vs. 130 Hz and targeted regions), this is still not able to fully explain why studies using similar paradigms, targeting somewhat analogous regions, reached different conclusions. As for the LH, initial studies in rodents [55,56] and felines [57] have largely shown that stimulation induced feeding. Part of those findings, however, has been attributed to the stimulation of structures adjacent to the LH, such as the medial forebrain bundle [27]. The MFB is comprised of axonal projections that interconnect over 50 brainstem, subcortical and cortical regions, including those involved in mechanisms of reward (e.g., LH, nucleus accumbens and ventral tegmental area). In contrast to earlier reports, however, recent studies in rodents using high frequency stimulation (e.g., above 100 Hz) have shown that LH DBS may reduce weight gain over time [58,59].

In contrast to the long history of hypothalamic stimulation for obesity, studies in which the nucleus accumbens was targeted in animal models are much more recent. Lesions of the NAc in rodents decrease food-hoarding behavior and are associated with weight loss [60]. Stimulation of the NAc shell delivered for 14 days to diet-induced obese rats led to significant reductions in total energy intake and weight gain, an effect that was associated with an up-regulation of the D2 receptor and increased DA levels [61]. Mice treated with NAc shell DBS were found to have a decrease in binge eating and an increase in immediate early gene expression in this same region [62]. D2 receptor antagonists attenuated DBS effects. In diet-induced obese mice, chronic NAc shell DBS has been found to reduce caloric intake and lead to weight loss. In rodents, stimulation of the accumbens core has also been shown to reduce binge eating [63].

## 4. Clinical Studies

Stereotactic ablative surgery targeting the hypothalamus for the treatment of obesity was initially been carried 40 years ago [64,65]. Overall, the procedure was proven to be safe, resulting in significant, though transient (e.g., few weeks), reductions in appetite and weight loss.

In 2008, Hamani et al. reported on a single patient with obesity treated with hypothalamic DBS [17]. Postoperative reconstruction of the electrode placement has shown that contacts used for stimulation were located near the fornix. While no weight changes were observed with high frequency DBS (130 Hz), when stimulated at 3.0–4.0 V, 210 μsec and 50 Hz, the patient lost 12 kg over five months. This was not due to significant dietary changes but to a reduction in food craving [17]. Over time, however, the patient reported that he was unable to sleep with the system activated and started turning it off at night. Without stimulation, he began nighttime binging and regained the weight he had lost [17].

In a more recent trial, Whiting et al. reported on the safety, efficacy and calorimetric effects of LH DBS in patients with obesity followed for an average of 35 months [18]. The three individuals in that study were diagnosed with refractory morbid obesity, which included a failure of bariatric surgery. Several scales were applied to assess the effects of DBS on eating and quality of life, including the Gormally Binge Eating Scale, the Cognitive Restraint subscale (used to assess dieting skills), a hunger scale, the Body Shape Questionnaire, and the Impact of Weight on Quality of Life–Lite Questionnaire. Though one of the participants had postoperative improvements in some of these scales, overall testing suggested that DBS did not induce significant changes. Also unchanged after DBS were blood tests to measure nutritional status, pituitary hormones, and neuroendocrine/neuropeptide studies. The most striking aspect of the trial was that DBS significantly increased resting metabolic state (RMR) in two patients. In these subjects, RMR improvement was in the order of 28% and 9%. Despite this fact, no consistent weight changes were noticed. DBS settings were 90 μsec, 185 Hz at different stimulation voltages [18].

In 2016, Harat and colleagues implanted NAc DBS electrodes in a 19-year-old patient who developed hypothalamic obesity following the onset and surgery for the removal of a craniopharyngioma [19]. Her weight before surgery was 151.4 kg. Three months after DBS she weighed 132 kg. Over time, her weight fluctuated due to a few instances in which the pacemaker was accidentally switched off. At the last follow-up visit (14 months after DBS), her weight was 138 kg. During periods in which the electrodes were found to be off, she reported increased food craving. This is of importance, as reductions in food craving and compulsive eating are some of the mechanisms through which NAc DBS has been postulated to exert its effects. Settings used in that study were 2–3.75 mA, 130 Hz and 208 μsec [19].

## 5. Future Perspectives and Applications of DBS in Obesity

To date, experience with DBS for obesity is quite limited. Overall, surgery has been proven to be safe, but no definitive conclusions can be made as to whether it is effective. Despite the number of studies published in animal models, several clinical aspects are still unclear, including target selection, the kinetics of DBS, or ideal stimulation parameters. Also unknown is whether DBS will work on genetic forms of obesity.

Prader Willi Syndrome (PWS) is one of the main causes of genetic obesity during childhood. Approximately 70% of patients have a deletion in chromosome 15 (15q11–q13) [66,67]. Of the remainder of patients, 25% have maternal uniparental disomy and 5% have imprinting defects. In infants and children, PWS is characterized by hypotonia, delayed neuropsychological development, lower-than-expected growth, hypogonadism, and hyperphagia [68,69]. Severe to morbid obesity is the most relevant problem of the syndrome due to associated comorbidities and early mortality. Hyperphagia in PWS is often refractory to pharmacological and psychotherapeutic approaches, as well as bariatric surgery [70]. Several factors suggest that the increased appetite in PWS may be associated with hypothalamic dysfunction [71]. First, the disease is comprised of a spectrum of hormonal problems (e.g., low levels of growth hormone, hypogonadotropic hypogonadism, temperature dysregulation). Post-mortem studies in patients with PWS have shown a reduced number of cells in the paraventricular nucleus, including neurons that synthesize anorexigenic hormones and oxytocin [72]. In PWS, imaging studies have shown an increased activation of reward circuits when patients are presented with food stimuli [73]. Bearing these facts in mind, DBS in either the hypothalamus or NAc has been hypothesized as a suitable alternative for the treatment of patients with PWS [74,75].

Another aspect that deserves to be discussed is the advancement of DBS technology, which may improve the delivery of stimulation to some of the targets discussed above. The hypothalamus is a small structure responsible for modulating various physiological functions. In this target, directional leads could certainly be of help to steer current into specific regions while avoiding spread to adjacent undesirable structures [76,77]. This may, in theory, improve safety and reduce the incidence of stimulation-induced side effects.

## 6. Conclusions

Data from animal studies and preliminary reports in humans suggest that DBS may be a promising alternative for the treatment of obesity. Structures involved in mechanisms of feeding and rewarding aspects of food intake, namely the hypothalamus and NAc, have been considered as potential targets. Though surgery was shown to be safe in the few patients treated so far, further studies are still needed not only to better characterize the side effect profile of these procedures but also their actual efficacy.

As in previous functional neurosurgery studies, trials on DBS for obesity have to take several issues into account. Inclusion criteria should be strict, with a clearly defined diagnosis, measures of refractoriness and recruitment of patients with severe forms of the disease. In addition, trials need to be carried out in an ethical manner [78,79,80], with particular attention paid to the informed consent process, long-term follow up, clinical care and support. Ideally, patients should be assessed by a multidisciplinary team of endocrinologists, neurosurgeons, psychiatrists and neuropsychologists, so that comprehensive care may be provided.

In summary, results of preclinical and early clinical trials using DBS for obesity have been promising. However, numerous questions remain unanswered, including the optimal target, stimulation parameters, and clinical aspects of the patients to be included (e.g., previous failure to bariatric surgery, compulsive eating, among other). Further work is certainly needed to address these issues.

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
