# Peer review of "DBS for Obesity"

_brainsci, 2016, doi:10.3390/brainsci6030021_

Round 1
Reviewer 1 Report
Excellent and timely review. Up to date, comprehensive bibliography
Author Response
We would like to thank this reviewer for the positive evaluation.
Reviewer 2 Report
First paragraph of the introduction would benefit from more updated statistics on the global obesity burden (ie, 2014 or more recent).
Second paragraph, third sentence of introduction the word ‘prematurerely’ should be spelled ‘prematurely’.
Third paragraph, third sentence of introduction the word ‘nowadays’ in colloquial and should be replaced.
Introduction would benefit from some more in-depth discussion of the pivotal bariatric surgery studies with significant long-term recurrence rates, and the evidence within the bariatric surgery literature that bolsters the point that there exists a significant compulsive eating/neuropsychiatric component to bariatric surgery failures.
In the description of DBS in the fourth paragraph of the introduction, can be more vague about the contacts and electrode configurations as there are many different types currently in use.
In the first paragraph, second sentence of Anatomical Targets, ‘accumbes’ should be spelled ‘accumbens’.
In the second paragraph, fourth sentence of Anatomical Targets, ‘neuropsptide’ should be spelled ‘neuropeptide’.
The second paragraph of Anatomical Targets should include more discussion of the pathophysiology of the hypothalamus in feeding and satiety (ie, more basic discussion).
The third paragraph of Anatomic Targets begins ‘Multiple evidence suggest’ which is awkward word choice to begin this sentence, please rewrite for clarity.
Again, the third paragraph of Anatomic Targets should include more discussion of the pathophysiology of the nucleus accumbens in addiction behavior (ie, more basic discussion, what are the D2 receptors and how are they associated with the NAc)
Also, in the third paragraph, last sentence they cite a study of obese patients treated with bariatric surgery and their NAc response to food, please expand on this and how it bolsters the argument that NAc should be a DBS obesity target.
First paragraph, first sentence of Preclinical Studies, would consider rewording this sentence to be more of an introductory sentence to preclinical studies for the targets discussed.
First paragraph, third sentence of Preclinical Studies should read: “…have shown to increase feeding or satiety, respectively.”
In the first paragraph of Preclinical Studies, would recommend selecting a few pivotal studies of VM stimulation and discussing in depth (both positive and negative studies), rather than citing multiple studies in different animals and discussing broad strokes their aggregated results. This will help the reader understand the nuances of these studies (stim parameters, animal models, targeting) that may help explain the disparate results in the literature.
Again, towards the end of the first paragraph of Preclinical Studies during the discussion of LH stimulation, would discuss in more detail the rodent study with high-frequency stimulation that did reduce weight gain and why those results differed from previous studies that demonstrated an increase in feeding behavior.
Second paragraph of Preclinical Studies will need to pathophysiology of the NAc role in addiction and linkage to food binging behavior.
First paragraph, second sentence of the Future perspectives section, ‘wehther’ should be spelled ‘whether’.
First paragraph, third sentence of the Future perspective section, ‘unkwon’ should spelled ‘unknown’.
First paragraph, line 151 of Future perspectives section should read “…and 5% have imprinting defects.”
First paragraph, line 155 of Future perspectives section, ‘psychotherapic’ should be ‘psychotherapeutic’
First paragraph, line 156 of Future perspectives section, ‘apetite’ should be ‘appetite’
First paragraph, line 163 of Future perspectives section, ‘hypothalamuc’ should be ‘hypothalamus’
Please cite the following papers in your Future perspectives section discussing PWS since they first propose the concept of applying DBS to treating obesity and compulsive behavior in PWS:
Ho AL, Sussman ES, Pendharkar AV, Azagury DE, Bohon C, Halpern CH. Deep Brain Stimulation for Obesity: Indications and Patient Selection. Neurosurg Focus. 2015 Jun;38(6):E8. doi: 10.3171/2015.3.FOCUS1538.
AL Ho, ES Sussman, AV Pendharkar, DE Azagury, C Bohon, CH Halpern. Deep Brain Stimulation for Obesity. Cureus 7(3): e259. doi:10.7759/cureus.259
First sentence of second paragraph of the Future perspectives section should read “…is the advancement of DBS technology…”
Line 168 of second paragraph of the Future perspectives section, ‘curent’ should be ‘current’
Line 169 of second paragraph of the Future perspectives section, ‘stimualtion’ should be ‘stimulation’
In the second paragraph of the Future perspectives section, please expand on your discussion of advances in DBS techniques that could be applied towards obesity targets. Please cite relevant papers/studies detailing these advances.
Line 173, ‘treamtemnt’ should be ‘treatment’
Line 174 should read “…with clearly defined diagnosis, measures of refractoriness, and inclusion…”
Line 176, ‘caracteristics’ should be ‘characteristics’
Line 177, ‘multidisciplinar’ should be ‘multidisciplinary’
Please rewrite Conclusion section to better recapitulate the points covered before in the paper.
Author Response
First paragraph of the introduction would benefit from more updated statistics on the global obesity burden (ie, 2014 or more recent).
A more updated statistic with WHO data was added to the article.
Second paragraph, third sentence of introduction the word ‘prematurerely’ should be spelled ‘prematurely’.
This has now been corrected.
Third paragraph, third sentence of introduction the word ‘nowadays’ in colloquial and should be replaced.
Nowadays was replaced by currently.
Introduction would benefit from some more in-depth discussion of the pivotal bariatric surgery studies with significant long-term recurrence rates, and the evidence within the bariatric surgery literature that bolsters the point that there exists a significant compulsive eating/neuropsychiatric component to bariatric surgery failures.
A sentence addressing the failure of bariatric surgery and its potential association with compulsive eating was added to the text.
In the description of DBS in the fourth paragraph of the introduction, can be more vague about the contacts and electrode configurations as there are many different types currently in use.
We agree that there are many novel electrodes and systems under investigation. However, we wanted to give clinicians potentially interested in the article a general overview of the settings/systems most commonly used. To address the comment, we have changed the sentence, which now reads as follows: “The electrodes most commonly used today have four different contacts, which may be activated…”
In the first paragraph, second sentence of Anatomical Targets, ‘accumbes’ should be spelled ‘accumbens’.
This has now been corrected.
In the second paragraph, fourth sentence of Anatomical Targets, ‘neuropsptide’ should be spelled ‘neuropeptide’.
This has now been corrected.
The second paragraph of Anatomical Targets should include more discussion of the pathophysiology of the hypothalamus in feeding and satiety (ie, more basic discussion).
The above-mentioned paragraph has been rewritten and now includes additional data on the role of the hypothalamus in mechanisms of feeding.
The third paragraph of Anatomic Targets begins ‘Multiple evidence suggest’ which is awkward word choice to begin this sentence, please rewrite for clarity.
This sentence has been rewritten.
Again, the third paragraph of Anatomic Targets should include more discussion of the pathophysiology of the nucleus accumbens in addiction behavior (ie, more basic discussion, what are the D2 receptors and how are they associated with the NAc)
We appreciate the comment. Instead of incorporating a section on the role of NAc in mechanisms of addiction, which would be beyond the scope of this review, we now provide a few extra statements on neuroimaging data in obesity, including a decrease in D2 binding (last paragraph section 2).
Also, in the third paragraph, last sentence they cite a study of obese patients treated with bariatric surgery and their NAc response to food, please expand on this and how it bolsters the argument that NAc should be a DBS obesity target.
The following statement was added to the paragraph: “This is of importance as reductions in food craving and compulsive eating are some of the mechanisms through which NAc DBS has been postulated to exert its effects”.
First paragraph, first sentence of Preclinical Studies, would consider rewording this sentence to be more of an introductory sentence to preclinical studies for the targets discussed.
The initial sentences of the above-mentioned paragraph were rewritten, as suggested.
First paragraph, third sentence of Preclinical Studies should read: “…have shown to increase feeding or satiety, respectively.”
This no longer applies, as the sentence was rewritten.
In the first paragraph of Preclinical Studies, would recommend selecting a few pivotal studies of VM stimulation and discussing in depth (both positive and negative studies), rather than citing multiple studies in different animals and discussing broad strokes their aggregated results. This will help the reader understand the nuances of these studies (stim parameters, animal models, targeting) that may help explain the disparate results in the literature.
Again, towards the end of the first paragraph of Preclinical Studies during the discussion of LH stimulation, would discuss in more detail the rodent study with high-frequency stimulation that did reduce weight gain and why those results differed from previous studies that demonstrated an increase in feeding behavior.
Second paragraph of Preclinical Studies will need to pathophysiology of the NAc role in addiction and linkage to food binging behavior.
We appreciate the comment. Yet, in our view providing more general statements would be more informative to the readers than reporting a few studies in further detail. Our goal was to present an overview of the field and not an extensive description of preclinical data.
First paragraph, second sentence of the Future perspectives section, ‘wehther’ should be spelled ‘whether’.
This has now been corrected.
First paragraph, third sentence of the Future perspective section, ‘unkwon’ should spelled ‘unknown’.
This has now been corrected.
First paragraph, line 151 of Future perspectives section should read “…and 5% have imprinting defects.”
This has now been corrected.
First paragraph, line 155 of Future perspectives section, ‘psychotherapic’ should be ‘psychotherapeutic’
This has now been corrected.
First paragraph, line 156 of Future perspectives section, ‘apetite’ should be ‘appetite’
This has now been corrected.
First paragraph, line 163 of Future perspectives section, ‘hypothalamuc’ should be ‘hypothalamus’
This has now been corrected.
Please cite the following papers in your Future perspectives section discussing PWS since they first propose the concept of applying DBS to treating obesity and compulsive behavior in PWS:
Ho AL, Sussman ES, Pendharkar AV, Azagury DE, Bohon C, Halpern CH. Deep Brain Stimulation for Obesity: Indications and Patient Selection. Neurosurg Focus. 2015 Jun;38(6):E8. doi: 10.3171/2015.3.FOCUS1538.
AL Ho, ES Sussman, AV Pendharkar, DE Azagury, C Bohon, CH Halpern. Deep Brain Stimulation for Obesity. Cureus 7(3): e259. doi:10.7759/cureus.259
References by Ho et al were added to the text.
First sentence of second paragraph of the Future perspectives section should read “…is the advancement of DBS technology…”
This has now been corrected.
Line 168 of second paragraph of the Future perspectives section, ‘curent’ should be ‘current’
This has now been corrected.
Line 169 of second paragraph of the Future perspectives section, ‘stimualtion’ should be ‘stimulation’
This has now been corrected.
In the second paragraph of the Future perspectives section, please expand on your discussion of advances in DBS techniques that could be applied towards obesity targets. Please cite relevant papers/studies detailing these advances.
Papers on new technologies have now been quoted.
Line 173, ‘treamtemnt’ should be ‘treatment’
Line 174 should read “…with clearly defined diagnosis, measures of refractoriness, and inclusion…”
Line 176, ‘caracteristics’ should be ‘characteristics’
Line 177, ‘multidisciplinar’ should be ‘multidisciplinary’
The above-mentioned typographical errors have now been corrected.
Please rewrite Conclusion section to better recapitulate the points covered before in the paper.
The conclusions have been partly rewritten, including a few sentences of ethical considerations, which were requested by reviewer 3.
Reviewer 3 Report
Page | Line | Comment |
1 | 18 | change "but associated" to "but is associated" |
1 | 31 | Change "8" to "eight" |
1 | 33 | change "but also" to "but they also" |
1 | 38 | Could you mention what those side effects are? |
1 | 43 - 44 | Reference for "One of the main factors associated with disease recurrence is compulsive eating" |
1 | 45 | Remove comma between "condition" and "but" |
2 | 59 | Could you provide a short clause after "accumbens" that briefly describes which behaviors are regulated by the hypothalamus and nucleus accumbens? |
2 | 60 - 68 | Could you provide some information on the involvement of LH and VM in feeding and satiety from neural recordings in rodents and/or neuroimaging studies in humans? Also, is neural/ brain activity (glucose metabolism) in these regions at rest and/or during specific tasks altered in obese individuals? |
2 | 69 - 75 | Could you provide some information on the involvement of LH and VM in feeding and satiety from neural recordings in rodents, if available? |
2 | 77 | Insert a comma between "today" and "the" |
2 | 80 | Change "were" to "was" |
2 | 89 | Change "This, however, has not been uniform" to "However, these results are not consistent" |
2 | 93 | Perhaps, the LH part could be made into a new paragraph. |
3 | 77 - 110 | Maybe you could provide a table to summarize results from animal studies. |
3 | 112 | Change "was" to "has" |
3 | 114 | How long did the weight reduction and appetite loss persist? |
3 | 141 | Insert a comma between "off" and "she" |
3 | 130-131 | Change "blood tests to measure nutritional status, pituitary hormones, and neuroendocrine/neuropeptide studies" to "nutritional status, and blood pituitary hormone and neuroendocrine/ neuropeptide levels" |
4 | 146 | Change "wehther" to "whether" |
4 | 146 | Insert a comma between "safe" and "but" |
4 | 148 | Change "unkwon" to "unknown" |
4 | 145 - 148 | Perhaps you could further expand this, speculating on possible reasons for the not so convincing results from these studies. |
4 | 149 | Maybe, the discussion of PWS could be written as a separate paragraph |
4 | 151 | Change "uniparental dissomy and 5% imprinting" to "uniparental disomy, and 5% have imprinting" |
4 | 165 | Change "advance" to "advancements" |
4 | 165 - 170 | How about the nucleus accumbens? Will directional leads also provide an advantage? |
4 | 173 | Change "treamtemnt" to "treatment" |
4 | 175-176 | Reference for "Obese patients with these caracteristics often have an increased surgical risk" |
4 | 176 | Change "caracteristics" to "characteristics" |
4 | 177 | Change "multidisciplinar" to "multidisciplinary" |
Overall
Overall
Overall | Methods
Safety & ethics
Reference | Maybe you could provide a short description on the parameters of the search used to obtain information for this review. Which databases were used, what are the search terms, and when was the search done?
We strongly urge authors to include, or at least, wave at potential safety and ethical issues raises by using implantable brain device to treat obesity. For instance, authors could include a short sentence referring to work done, for example, by Pisapia, J. M., C. H. Halpern, U. J. Muller, et al. 2013. Ethical considerations in deep brain stimulation for the treatment of addiction and overeating associated with obesity. AJOB Neuroscience 4(2): 35–46. Gilbert, F. 2012. The Burden of normality: From ‘chronically ill’ to ‘symptom free’. New ethical challenges for deep brain stimulation postoperative treatment. Journal of Medical Ethics 38: 408–412. doi:10.1136/medethics-2011-100044 Authors should double-check their references, for instance, reference line 188 seems inaccurate. |
Author Response
Authors
Page | Line | Comment |
1 | 18 | change "but associated" to "but is associated"
This has now been corrected.
|
1 | 31 | Change "8" to "eight"
This has now been corrected.
|
1 | 33 | change "but also" to "but they also"
This has now been corrected.
|
1 | 38 | Could you mention what those side effects are?
We hope the reviewer agrees that this would be an extensive topic, as complications of bariatric surgery may range from neurologic, metabolic, gastrointestinal, among many others. In addition, there is a whole pool of surgical/technical problems that may happen, in addition to infections, bleeding, etc. As such, we have decided to leave the text as was and just include a reference that interested readers may report to.
|
1 | 43 – 44 | Reference for "One of the main factors associated with disease recurrence is compulsive eating"
A reference was added to the sentence.
|
1 | 45 | Remove comma between "condition" and "but"
This has now been corrected.
|
2 | 59 | Could you provide a short clause after "accumbens" that briefly describes which behaviors are regulated by the hypothalamus and nucleus accumbens?
This has now been provided in section 2.
|
2 | 60 – 68 | Could you provide some information on the involvement of LH and VM in feeding and satiety from neural recordings in rodents and/or neuroimaging studies in humans? Also, is neural/ brain activity (glucose metabolism) in these regions at rest and/or during specific tasks altered in obese individuals?
Section 2 has been rewritten. We note, however, that electrophysiology data in rodents that may be of importance to understand mechanisms of DBS is not plenty. Studies have been mainly carried to ascertain the role of specific cells, circuits in mechanisms of feeding. We now provide a few lines on neuroimaging study results in obese patients.
|
2 | 69 – 75 | Could you provide some information on the involvement of LH and VM in feeding and satiety from neural recordings in rodents, if available?
As described above, most electrophysiological data was carried out to ascertain the role of specific cells/ peptides on mechanisms of feeding behavior. Adding this sort of information would require us to describe in detail cells, transmitters and substrates involved in feeding, which would be beyond the scope of this review.
|
2 | 77 | Insert a comma between "today" and "the"
This has now been corrected.
|
2 | 80 | Change "were" to "was"
This has now been corrected.
|
2 | 89 | Change "This, however, has not been uniform" to "However, these results are not consistent"
This has now been corrected.
|
2 | 93 | Perhaps, the LH part could be made into a new paragraph.
Some of the sections including LH data have been rewritten.
|
3 | 77 - 110 | Maybe you could provide a table to summarize results from animal studies.
There are dozens of animal studies published to data. Summarizing the results in a single table would not only be cumbersome but we would run the risk of not reporting all the studies published to date. We hope the reviewer agrees that this would not be practical for an general review on DBS and obesity.
|
3 | 112 | Change "was" to "has"
This has now been corrected.
|
3 | 114 | How long did the weight reduction and appetite loss persist?
After lesions, weight loss was more pronounced and lasted only during the first few postoperative weeks. This has now been mentioned in the text.
|
3 | 141 | Insert a comma between "off" and "she"
This has now been corrected.
|
3 | 130-131 | Change "blood tests to measure nutritional status, pituitary hormones, and neuroendocrine/neuropeptide studies" to "nutritional status, and blood pituitary hormone and neuroendocrine/ neuropeptide levels"
This change would modify the structure of the sentence. “Blood tests to measure nutritional status” is not equivalent to “nutritional status”.
|
4 | 146 | Change "wehther" to "whether"
This has now been corrected.
|
4 | 146 | Insert a comma between "safe" and "but"
This has now been corrected.
|
4 | 148 | Change "unkwon" to "unknown"
This has now been corrected.
|
4 | 145 - 148 | Perhaps you could further expand this, speculating on possible reasons for the not so convincing results from these studies.
We appreciate the comment. In fact, we have thought about it. At the moment, however, none of the investigators could raise plausible hypotheses as to why results have not been striking. As such, we did not discuss potential reasons for the unconvincing findings published to date. We hope the reviewer understands that any attempt would be purely speculative.
|
4 | 149 | Maybe, the discussion of PWS could be written as a separate paragraph.
PWS is now discussed in a different paragraph, as suggested.
|
4 | 151 | Change "uniparental dissomy and 5% imprinting" to "uniparental disomy, and 5% have imprinting"
This has now been corrected.
|
4 | 165 | Change "advance" to "advancements"
This has now been corrected.
|
4 | 165 - 170 | How about the nucleus accumbens? Will directional leads also provide an advantage?
In the NAc, this is less clear since it is a relatively larger structure. In this context, we did not include the NAc as the example of a structure in which directional leads would be of significant value.
|
4 | 173 | Change "treamtemnt" to "treatment"
This has now been corrected.
|
4 | 175-176 | Reference for "Obese patients with these caracteristics often have an increased surgical risk"
This paragraph was changed and this sentence was removed.
|
4 | 176 | Change "caracteristics" to "characteristics"
This word was removed from the text.
|
4 | 177 | Change "multidisciplinar" to "multidisciplinary"
This has now been corrected.
|
Overall
Overall
Overall | Methods
Safety & ethics
Reference | Maybe you could provide a short description on the parameters of the search used to obtain information for this review. Which databases were used, what are the search terms, and when was the search done?
We did not apply search terms. We looked for articles on PubMed. Only those we deemed to be of interest were quoted, except for the clinical section, in which all trials published to date are mentioned.
We strongly urge authors to include, or at least, wave at potential safety and ethical issues raises by using implantable brain device to treat obesity.
For instance, authors could include a short sentence referring to work done, for example, by Pisapia, J. M., C. H. Halpern, U. J. Muller, et al. 2013. Ethical considerations in deep brain stimulation for the treatment of addiction and overeating associated with obesity. AJOB Neuroscience 4(2): 35–46. Gilbert, F. 2012. The Burden of normality: From ‘chronically ill’ to ‘symptom free’. New ethical challenges for deep brain stimulation postoperative treatment. Journal of Medical Ethics 38: 408–412. doi:10.1136/medethics-2011-100044
A few lines were added on safety and ethical aspects, as suggested. The references by Pisapia and colleagues and Gilbert are now quoted.
Authors should double-check their references, for instance, reference line 188 seems inaccurate.
The reviewer is correct. Our first reference had a problem. |
Round 2
Reviewer 2 Report
All requested changes were made.